# Quantitation of MicroRNA-155 in Human Cells by Heterogeneous Enzyme-Linked Oligonucleotide Assay Coupled with Mismatched Catalytic Hairpin Assembly Reaction

**DOI:** 10.3390/bios12080570

**Published:** 2022-07-26

**Authors:** Oleg L. Bodulev, Ivan I. Galkin, Shulin Zhao, Olga Y. Pletyushkina, Ivan Y. Sakharov

**Affiliations:** 1Department of Chemistry, Lomonosov Moscow State University, Leninskie Gory, Bldg. 1, 119991 Moscow, Russia; bodulevoleg@mail.ru; 2A.N. Belozersky Research Institute of Physico-Chemical Biology, Lomonosov Moscow State University, Leninskie Gory, Bldg. 1, 119992 Moscow, Russia; ddeathseller@gmail.com (I.I.G.); pletjush@genebee.msu.ru (O.Y.P.); 3State Key Laboratory for the Chemistry and Molecular Engineering of Medicinal Resources, Guangxi Normal University, Guilin 541004, China; zhaoshulin001@163.com

**Keywords:** microRNA detection, polymerase-free isothermal amplification of nucleic acids, mismatched catalytic hairpin assembly

## Abstract

In the present work, we describe the development of a chemiluminescent enzyme-linked oligonucleotide assay coupled with mismatched catalytic hairpin assembly (mCHA) amplification for the quantitative determination of microRNA-155. To improve its sensitivity, a polymerase-free mCHA reaction was applied as an isothermal amplification method. The detection limit of the proposed assay was 400 fM. In addition, the high specificity of the assay was demonstrated. The proposed assay allowed assessment of the content of microRNA-155 in human cancer lines such as HepG2, Caco2, MCF7, and HeLa. The quantitation of microRNA-155 was performed after purification of short RNAs (less than 200 nt) from cell lysates since a high matrix effect was observed without this pre-treatment. The results of the quantitative determination of the microRNA content in cells were normalized using nematode microRNA-39, the concentration of which was determined using a heterogeneous assay developed by us using a strategy identical to that of the microRNA-155 assay.

## 1. Introduction

MicroRNAs (miRNAs) are short (17–25 nucleotides in length) non-coding RNA oligonucleotides that regulate gene expression by interacting with specific sites of messenger RNA causing translational repression or its degradation [1,2]. MiRNAs were established to be involved in the regulation of important biological processes such as the development, differentiation, metabolism, and the formation of an immunological response [3]. Currently, about 2600 miRNAs have been detected in the human organism [4]. In addition, previous studies have shown that the miRNA content in tissues, cells, and biological fluids undergo changes in a wide range of diseases [5]. In particular, a relationship was observed between fluctuations in the miRNA expression levels and the development of human oncological diseases. A comparative analysis of samples of healthy individuals and cancer patients revealed upregulation or downregulation of the expression of some miRNAs [6]. This effect depended on the type and stage of the disease. Moreover, a change in the concentration of miRNA was also noted during treatment of patients. Considering the above, numerous efforts are being carried out to develop analytical methods for the determination of miRNA biomarkers.

It should be noted that the concentration of miRNAs in biological tissues and liquids is extremely low (in the fM–pM range [7,8]) and their determination in clinical samples is a challenge for analytical chemists. This means that practically significant miRNA methods should have a low detection limit and high sensitivity coefficient. Often, this is achieved by introducing an amplification stage into the analysis scheme [9,10]. Presently, the most used method of amplification is the reverse transcription polymerase chain reaction (RT-PCR), which is widely applied to copy miRNA sequences. However, RT-PCR has some disadvantages associated with the need to vary the temperature during the analysis, which requires the use of expensive equipment. In addition, miRNAs are short oligonucleotides, and they must be extended in RT-PCR.

These shortcomings have forced researchers to actively look for alternative methods for the amplification of miRNAs. Currently, the prospects of using isothermal methods of miRNA amplification have been demonstrated [11,12,13]. Among them, a special niche is occupied by enzyme-free methods since they are cheaper. Catalytic hairpin assembly (CHA) and its improved variant, called mismatched CHA (mCHA), are most commonly used.

Another important parameter of miRNA assays is their specificity. This is due to the fact that the number of known miRNAs is high, and their sequences are short, i.e., miRNAs show high homology. Therefore, only highly sensitive and highly specific miRNA assays can be useful for their practical application.

One of the miRNAs considered as a promising biomarker in early diagnosis of cancer is miRNA-155 [5,14]. Its concentration was found to increase in the serum of breast cancer patients [15]. Overexpression of miRNA-155 significantly promotes the proliferation of breast cancer cells, which was explained by the silencing of the Socs1 and RhoA tumor-suppressing genes [16]. Interestingly, the microRNA-155 concentration was reduced after surgery or chemotherapy in patients with breast cancer [17], i.e., the concentration of miRNA-155 can be used to assess the condition of patients during treatment.

In this work, a chemiluminescent heterogeneous method for the determination of miRNA-155 was developed. To enhance the assay’s sensitivity, mCHA reaction was used as a polymerase-free amplification method. Furthermore, the use of streptavidin-polyperoxidase conjugate and an enhanced chemiluminescence reaction additionally amplified the chemiluminescent signal. The proposed assay made it possible to quantify miRNA-155 in cancer cell lines such as HeLa, HepG2, Caco2, and MCF-7.

## 2. Experimental

### 2.1. Reagents, Materials, and Equipment

Milk casein, Tween-20, 4-morpholinopyridine, Tris, and luminol were obtained from Sigma-Aldrich (St. Louis, MO, USA). Monoclonal anti-fluorescein antibody (anti-FluAb) belonging to the IgG class was produced by Bialexa (Moscow, Russia). Further, 30% H_2_O_2_ was purchased from ChimMed (Moscow, Russia). Chemiluminescent enhancer for HRP, 3-(10′-phenothiazinyl)-propionic acid was produced as described in [18]. Conjugate of streptavidin and horseradish polyperoxidase (Str–PolyHRP80) was obtained from SDT GmbH (Baesweiler, Germany). Black multiwell microplates (High Binding) were obtained from Thermo Fisher Scientific (Waltham, MA, USA). All oligonucleotides used in this work (Appendix A) were obtained from Syntol (Moscow, Russia).

The human cells (MCF-7, HeLa, Caco2, and HepG2) were obtained from the Russian Collection of Cell Cultures (Institute of Cytology, St. Petersburg, Russia). All cells were cultured in Dulbecco’s Modified Eagle’s Medium with 10% fetal bovine serum at 37 °C in an atmosphere containing 5% CO_2_ as described in our previous work [19].

### 2.2. Measurement of MicroRNA Concentration

First, 96-well black microplates with adsorbed anti-FluAb were prepared as described in [20]. For this, 50 μL of 6 µg/mL monoclonal anti-FluAb in 50 mM carbonate buffer, pH 9.5 was added to the plate wells and incubated for 20 h at 4 °C. Then, the plates were rinsed three times with 10 mM Tris-HCl, pH 7.2 (TB) supplemented with 300 mM NaCl (TBS) and 50 μL of 1 mg/mL casein in TBS was added to the wells. After incubation at 37 °C for 60 min, the plates were rinsed three times with TBS supplemented with 0.05% Tween X100 (TBST). The prepared plates were stored at 4 °C.

Before being used in the assay, Flu-HP1 and B-HP2 were annealed at 88 °C for 15 min and cooled to room temperature for 1 h. The Flu-HP1 probe was annealed at a concentration of 30 nM in 10 mM Tris-HCl, pH 7.2 (TB) supplemented with 20 mM MgCl_2_. The B-HP2 probe was annealed at a concentration of 100 nM in TB supplemented with 10 mM MgCl_2_ [21]. The determination of the miRNA concentration in the samples was carried out as follows: In total, 50 μL of Flu-HP1 (30 nM in TB supplemented with 20 mM MgCl_2_) was added to the plate wells to bind with the adsorbed anti-FITC antibodies. After a 60-min incubation at 37 °C, the plates were rinsed three times with TBST. Afterward, 25 μL of aqueous solution of target miRNA (0–200 pM) or purified small RNA solution and 25 μL 160–1280 nM biotin-H2 in TB supplemented with 20–160 mM MgCl_2_ was added to the wells. The plates were incubated for 60 min at 25 °C and washed three times with TBST. Thereafter, the plates were filled with 50 μL of Str–PolyHRP80 conjugate solution (diluted 1:100,000 in TBS, supplemented with 1 mg/mL milk casein) and incubated for 60 min at 37 °C. The plates were rinsed thrice again with TBST and 100 μL of freshly prepared solution containing 100 mM Tris, pH 8.3, with 5.2 mM 3-(10′-phenothiazinyl) propionic acid, 9.3 mM 4-morpholinopyridine, and 1 mM luminol, and 3 mM H_2_O_2_ [22] was introduced to the wells. Chemiluminescence was analyzed using a SpectraMax L luminometer (Molecular Devices, Sunnyvale, CA, USA) at room temperature.

### 2.3. Purification and Quantitation of Cellular MicroRNAs

The purification of short RNAs from cell lysates was carried out using the LRU-100-50 kit from “BioLabMix” (Russia, Novosibirsk) as described previously [19]. Purified miRNA samples were stored at −20 °C. The quantitation of the microRNA-155 and microRNA-39 in the miRNA samples purified from the cell lysates was performed as described in Section 2.2.

## 3. Results and Discussion

### 3.1. Design of the MicroRNA Assay

The principle of the chemiluminescent enzyme-linked oligonucleotide assay for miRNA-155 determination is schematically presented in Figure 1. The mCHA reaction was employed for isothermal amplification of the analytical signal. The sequences of the hairpins used in this study were modeled according to the mCHA theory [23]. Their secondary structures are presented in Figure 2.

Immobilization of the HP1 hairpin probe labeled with fluorescein (Flu-HP1) occurred due to its interaction with the commercial anti-fluorescein antibody (anti-FluAb) pre-adsorbed on the surface of multiwell plates. Preliminarily, Flu-HP1 was annealed to form a hairpin structure [24]. During the immobilization of Flu-HP1, its concentration was 30 nM, since it was shown that this concentration allows the maximum chemiluminescent signal to be obtained [25].

In the presence of the analyte, the captured Flu-HP1 sequence containing an overhang located at the 3′ end and adjacent stem strand reacts with the miRNA-155 sequence, forming a primary duplex structure (Figure 3A). As a result, the HP1 region at the 5′ end becomes accessible due to its release from the stem. The released sequence then hybridizes with the protruding fragment of the biotin-labeled HP2 (B-HP2), resulting in the formation of the Flu-HP1/B-HP2 complex (Figure 3B). During the formation of the secondary complex, the miRNA-155 molecule is displaced by B-HP2 from the primary complex into the solution. Consequently, miRNA-155 is in a free state and is able to initiate the next cycle of mCHA amplification. The higher the number of amplification cycles, the more molecules of the Flu-HP1/B-HP2 complex are formed.

The amount of Flu-HP1/B-HP2 duplex produced was quantified by a reaction with the streptavidin-polyHRP conjugate, the enzyme activity of which was estimated by an enhanced chemiluminescence reaction [22]. The application of the triple amplification strategy led to a multiple increase in the intensity of the chemiluminescent signal.

In the absence of the target, the hairpins used that are in closed forms should not react with each other. Unfortunately, this reaction, depending on the assay conditions, can be observed to one degree or another. This results in the appearance of a background signal. To minimize the background, the experimental conditions for the mCHA-assisted assays should be optimized.

### 3.2. Optimization of the Assay Conditions for MicroRNA-155 Detection

As mCHA represents a catalytic interaction of hairpins, the main factor affecting the effectiveness of the amplification step in the mCHA-based assays and, therefore, its sensitivity is the concentration of hairpins used. The use of H1/H2 reactive hairpins at low concentrations slows down the reaction, which reduces the assay sensitivity. On the other hand, a sharp increase in the hairpin concentration also reduces the sensitivity due to high background. Thus, it is necessary to find the optimal concentration of hairpins in the reaction medium. It should be noted that the optimal concentrations of hairpins may also depend on the time of the amplification reaction.

In order for the proposed analysis to not take much time, the amplification was carried out for 1 h. The concentration of the B-HP2 probe in the reaction solution varied within 80–640 nM. Under these conditions, a set of calibration curves were constructed for the determination of miRNA-155 (Figure 4A). For each curve, the detection limit was calculated by the 3σ rule. As seen in Figure 4B, a minimum detection limit was obtained using the B-HP2 probe at a concentration of 160 nM. At other concentrations of B-HP2, the sensitivity of the assay was lower.

We also analyzed the dependence of the chemiluminescent signal formed in the presence of 100 pM of miRNA-155 on the concentration of the B-HP2 probe (Figure 4). When the concentration of the B-HP2 probe was in the range of 160–320 nM, a maximum signal was observed. The use of the B-HP2 probe outside this concentration range resulted in a decrease in the chemiluminescence. Considering the factors mentioned above, in the further work, we used the B-HP2 probe at a concentration of 160 nM.

It is well known that the salt composition of the reaction solution strongly affects the affinity of the interaction of complementary nucleic acids. NaCl and MgCl_2_ are salts that are usually used in the reactions of nucleic acids. The salt effect is especially significant when carrying out heterogeneous reactions [26]. In our work, we showed that the mCHA reaction of the immobilized Flu-H1 and soluble B-HP2 did not proceed in 10 mM Tris-HCl, pH 7.2. The introduction of MgCl_2_ to the reaction buffer intensified the mCHA and this effect was concentration dependent (Figure 5). The favorable concentration of MgCl_2_ was 20 mM. It should be noted that in 10 mM Tris-HCl, pH 7.2 containing 20 mM MgCl_2_, we observed not only the maximum chemiluminescent signal but also the maximum sensitivity (a minimum value of detection limit) for miRNA-155 detection (Figure 5). The addition of NaCl to the MgCl_2_-containing buffer had no positive effect on the assay sensitivity (data not shown).

### 3.3. Analytical Parameters of the Heterogenous mCHA-Based Assay of MicroRNA-155

Under favorable conditions (10 mM Tris-HCl of pH 7.2 with 20 mM MgCl_2_ and 160 nM B-HP2 probe), the behavior of the calibration curve for miRNA-155 detection (Figure 6) obeyed the following equation:$$\gamma =\frac{ax}{b+x}+\frac{cx}{d+x}\;(R^{2}\;0.9980)$$
where a, b, c and d are 2.5 × 10^6^, 0.22, 1.1 × 10^8^, and 1.06 × 10^3^, respectively (curve fitting was performed using SigmaPlot 12.5 software). The detection limit of miRNA-155 was 400 fM. The coefficient of variation of the chemiluminescent signal within the working range was less than 12%.

### 3.4. Specificity

To assess the specificity of the proposed assay, the cross-reactivity of target microRNA-155 and five widely studied miRNAs (miRNA-141, miRNA-319a, miRNA-21, miRNA-205, and miRNA-39) was investigated. As seen in Figure 7, only in the case of miRNA-155 a high chemiluminescent signal was observed.

For the other miRNAs studied, the signals were negligible. This result allowed us to characterize the developed method as highly specific.

### 3.5. Detection of MicroRNAs in Human Cells

The high matrix effect observed when analyzing samples with a complex composition, such as biological real samples, often does not allow a high sensitivity and the required accuracy to be obtained. In order to evaluate the matrix effect observed during the detection of miRNAs in cell lysates by the developed assay, we prepared spiked samples of miRNA-39 added to MCF-7 lysates. MiRNA-39 was chosen in this experiment because this nematode miRNA is absent in human tissues and fluids [27]. The determination of the concentration of miRNA-39 in the assay buffer and the spiked samples was carried out by the chemiluminescent heterogeneous assay, which was constructed using a strategy similar to that of the assay of miRNA-155 (Appendix A).

The results of the optimization of the experimental conditions of the miRNA-39 assay are presented in the Appendix A. The calibration curve of the heterogenous mCHA-based assay of miRNA-39 obeyed the following equation: $$\gamma =\frac{ax}{b+x}+\frac{cx}{d+x}\;(R^{2}\;0.9976)$$
where a, b, c and d are 5.03 × 10^6^, 6.5, 6.3 × 10^14^, and 5.8 × 10^9^, respectively (Appendix A). The detection limit and coefficient of variation of the chemiluminescent signal in the working range were 300 fM and less than 11%, respectively. Similar to the miRNA-155 assay, the assay of miRNA-39 was also highly specific (Appendix A).

When we tried to determine miRNA-39 in the crude MCF-7 lysate using the mCHA-based assay, we observed a very high matrix effect. Therefore, using the proposed assay, the direct quantitation of miRNA-39 cannot be carried out. To prevent this effect, a commercial kit LRU-100-50 (BioLabMix) for the purification of short RNAs (less than 200 nts in length), including miRNAs (see Section 2.3), was used. Actually, the matrix effect was not observed when the spiked samples were prepared by introducing miRNA-39 to a solution of short RNAs isolated from MCF-7 lysates (data not shown). The results obtained permitted us to conclude that the calibration curve generated when the analysis was performed in the assay buffer can be used to calculate the miRNA concentrations in pre-treated samples.

Although the BioLabMix kit prevents the matrix effect, its use results in a partial loss of miRNAs. To determine the purification yield, we prepared two spiked samples. One of the samples was prepared by adding miRNA-39 to the crude MCF-7 lysate followed by purification using the BioLabMix kit. Another sample was prepared by adding miRNA-39 to the solution of purified small RNAs prepared from the crude MCF-7 lysate with the same kit. The purification yield determined by comparing the chemiluminescent signals generated by the analysis of both spiked samples was in the range of 50–60%. Since miRNA molecules are highly homologous, we believe that the obtained values of the purification yield can be used to normalize the results of the detection of all miRNAs, including miRNA-155.

Using the BioLabMix kit, the content of miRNA-155 in some cultured human cells was evaluated by the proposed assay. The average content of miRNA-155 in HepG2 cells (cell line isolated from a hepatocellular carcinoma) was 180 ± 40 copies per cell. In Caco2 cells (epithelial cells isolated from colon cancer tissue), the content of the target was lower and equal to 68 ± 26 copies per cell. In contrast, in MCF7 cells (breast cancer cells) and HeLa cells (cell line isolated from a cervical carcinoma), miRNA-155 was not detected.

## 4. Conclusions

In the present work, we described the development of a chemiluminescent heterogeneous assay for the quantitative determination of miRNA-155. To improve its sensitivity, a polymerase-free mCHA was applied as an isothermal amplification method. This allowed the construction of an assay with a detection limit of 400 fM. The proposed assay also showed high specificity. It should be noted that the use of commercially available microplates as a carrier for the heterogeneous assay (in our work, we used 96-well plates, though 384- or 1536-well plates can also be used) makes it possible to easily standardize and automate the determination of miRNA-155.

The proposed assay allowed successful determination of the content of miRNA-155 in human cancer cell lines such as HepG2, Caco2, MCF7, and HeLa. The quantitation of miRNA-155 was performed after the purification of short RNAs (less than 200 nts) from cell lysates since without such pre-treatment, a high matrix effect was observed. The results of the evaluation of the miRNA content in cells were normalized using nematode miRNA-39, the concentration of which was determined using a heterogeneous assay developed by us using a strategy identical to that of the miRNA-155 assay.

## Figures and Tables

**Figure 1 biosensors-12-00570-f001:**
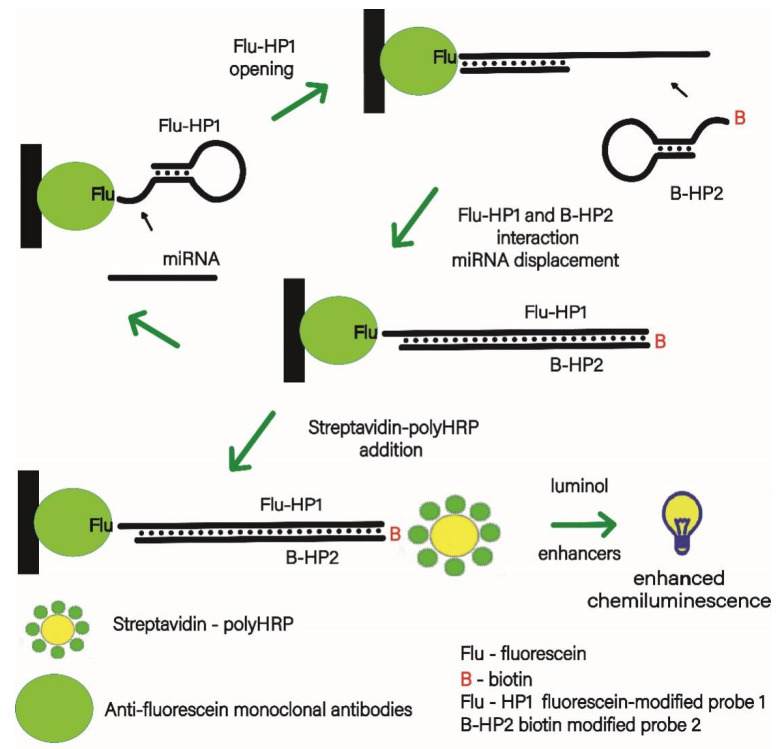
Scheme of the chemiluminescent heterogenous microRNA-155 assay coupled with mismatched catalytic hairpin assembly amplification.

**Figure 2 biosensors-12-00570-f002:**
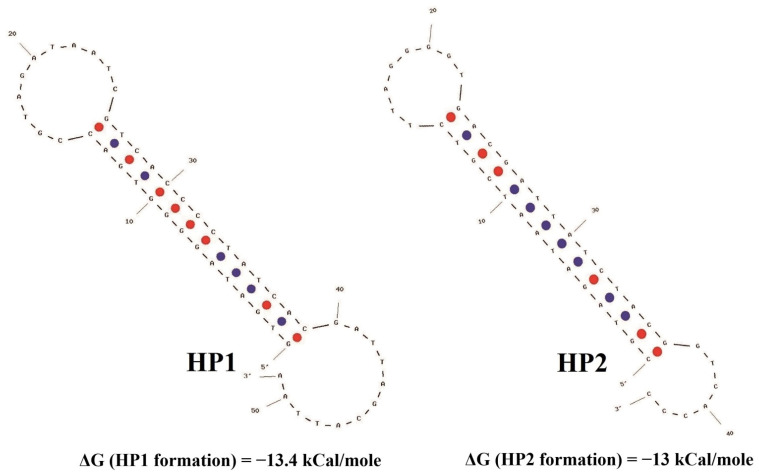
Secondary structure of the hairpins used in the chemiluminescent heterogenous assay based on the mismatched catalytic hairpin assembly reaction for microRNA-155 detection. Modeling of the hairpin structures and ΔG assessment were performed using OligoAnalyzer 3.1 software.

**Figure 3 biosensors-12-00570-f003:**
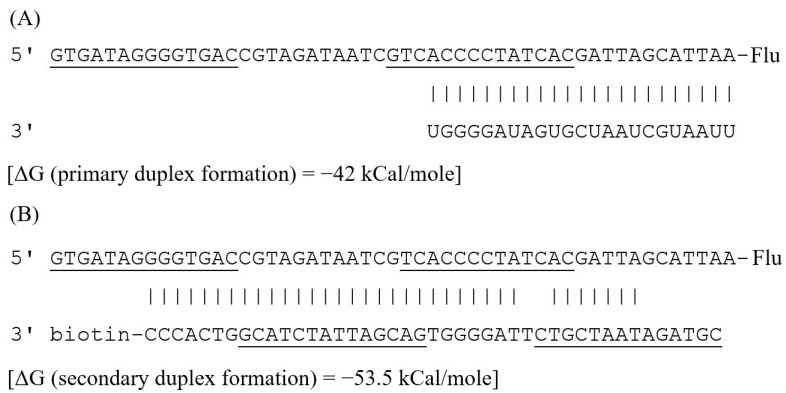
Schematic illustration of the formation of (**A**) primary and (**B**) secondary duplexes produced in the chemiluminescent assay based on the mismatched catalytic hairpin assembly reaction for microRNA-155 detection. The underlined nucleotide fragments are involved in the stems of the hairpin structures. Modeling of the hairpin structures and ΔG assessment were performed using OligoAnalyzer 3.1 software.

**Figure 4 biosensors-12-00570-f004:**
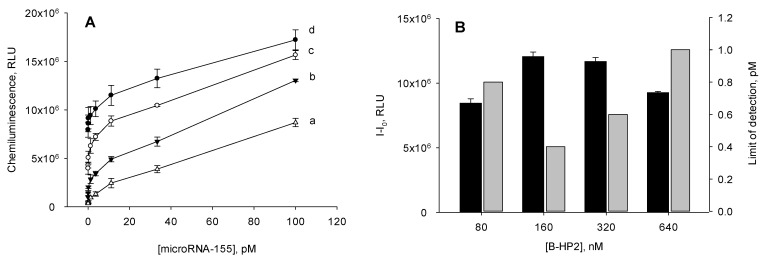
Effect of the B-HP2 probe concentration in the reaction solution on (**A**) the behavior of the calibration curves, (**B**) chemiluminescence signal (I–I_0_) (black columns), and limit of detection (gray columns) of the amplified microRNA-155 assay. The mismatched catalytic hairpin assembly reaction was carried out in 10 mM Tris-HCl with pH 7.2 containing 20 mM MgCl_2_ at 25 °C for 1 h. The concentration of the B-HP2 probe was (a) 80, (b) 160, (c) 320, and (d) 640 mM. The value of the chemiluminescence signal (I–I_0_) was calculated as the difference between the chemiluminescence intensities recorded in the presence (100 pM) and in the absence of microRNA-155. The detection limit was calculated using the 3σ rule.

**Figure 5 biosensors-12-00570-f005:**
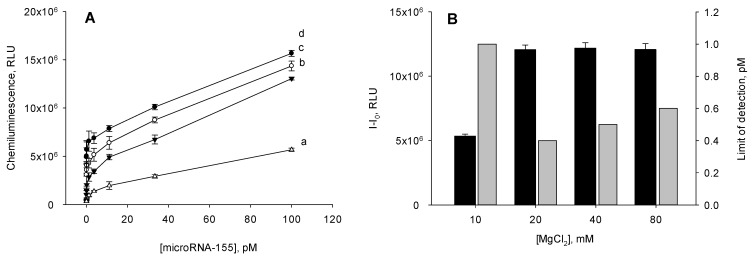
Effect of the MgCl_2_ concentration in the reaction solution at the stage of the mismatched catalytic hairpin assembly reaction (mCHA) amplification on (**A**) the behavior of the calibration curves constructed, (**B**) chemiluminescence signal (I–I_0_) (black columns), and limit of detection (gray columns) of the amplified microRNA-155 assay. The mCHA reaction was carried out at 25 °C for 1 h using the 160 nM B-HP2 probe in 10 mM Tris-HCl with pH 7.2 containing MgCl_2_ concentrations of (a) 10, (b) 20, (c) 40, and (d) 80 mM. The value of the chemiluminescence signal (I–I_0_) was calculated as the difference between the chemiluminescence intensities recorded in the presence (100 pM) and in the absence of the microRNA-155. The detection limit was calculated based on the 3σ rule.

**Figure 6 biosensors-12-00570-f006:**
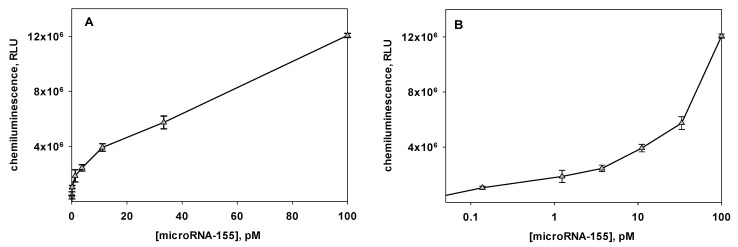
Calibration curve for the determination of microRNA-155 by the chemiluminescent heterogenous assay based on mismatched catalytic hairpin assembly amplification (n = 6) presented in (**A**) linear and (**B**) semi-logarithmic coordinates.

**Figure 7 biosensors-12-00570-f007:**
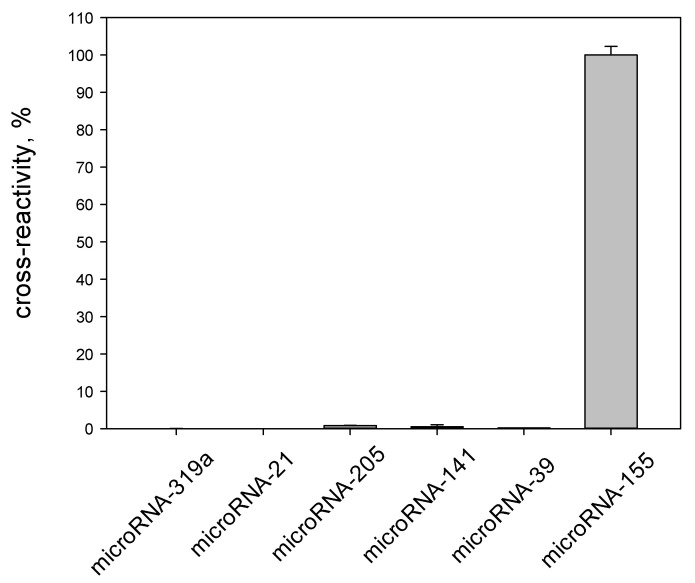
Specificity of the chemiluminescence heterogeneous method for the determination of microRNA-155 based on mismatched catalytic hairpin assembly amplification (n = 3). The concentration of the studied microRNAs was 100 pM.

## Data Availability

Not applicable.

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
