# Peer review of "Quantitation of MicroRNA-155 in Human Cells by Heterogeneous Enzyme-Linked Oligonucleotide Assay Coupled with Mismatched Catalytic Hairpin Assembly Reaction"

_biosensors, 2022, doi:10.3390/bios12080570_

Round 1
Reviewer 1 Report
In this manuscript, Bodulev et al. reported a chemiluminescent method to detect microRNA-155 using enzyme-linked oligonucleotides. Mismatched catalytic hairpin amplification was further introduced to improve the assay sensitivity. Moreover, the assay was used to detect miRNA in different cell lines. While the assay can provide sensitive and selective detection, the assay design is unnecessarily complicated, requiring multiple DNA probes and modified antibodies. Overall, the reviewer does not recommend this work for publication in Biosensors.
Author Response
We agree with the reviewer that the analytical method proposed by us is not simple. However, an analysis of the literature shows that simple assays of nucleic acids, including microRNAs, have low sensitivity and cannot be used in practice, since the concentration of nucleic acids in biological samples is low. Although it makes the analysis more complicate, nucleic acid amplification techniques are commonly used to increase sensitivity. The most popular nucleic acid amplification method is qRT-PCR. PCR-based kits are widely used in daily clinical practice for determination of nucleic acids. Since PCR has some disadvantages, isothermal amplification methods are currently used instead of PCR. One of these is the catalytic hairpin assembly (CHA) reaction and its more commonly used modification (mismatched catalytic hairpin assembly, mCHA). mCHA, which was used in our work, is a simple, cheap and polymerase-free amplification method. Thus, in our work we have developed the assay for the determination of miRNA-155, which, on the one hand, shows high sensitivity, and, on the other hand, is simpler and cheaper compared to other amplified miRNA assays. It should be noted that the proposed assay is optical and can be used as a screening assay. In addition, the use of microplates as a solid carrier will make it easy to automate the proposed analysis using ELISA instruments. Taking into account the above, we would like to note that our proposed assay is no more complicated than the microRNA assays described in the literature.
Reviewer 2 Report
(1) Any data that would indicate a mechanism such as the one shown in Fig. 1 should be presented. For example, data on hairpin structure, data on the basis of hybridization, etc.
(2) Is the sensitivity achieved this time sufficient?
Author Response
(1) Any data that would indicate a mechanism such as the one shown in Fig. 1 should be presented. For example, data on hairpin structure, data on the basis of hybridization, etc.
Accepted. We have described in more detail the principle of mCHA (see page 7 lines 153-159 and page 9 lines 181-186) and added Figure 2 with secondary structures of the used hairpins and Figure 3, which shows the reactions of HP1 and HP2 in the presence of microRNA-155.
(2) Is the sensitivity achieved this time sufficient?
As can be seen from our MS, the developed assay made it possible to evaluate the content of microRNA-155 in some human cancer cells. Thus, the obtained results allow to conclude that sensitivity of our assay is sufficient to solve such a task.
Reviewer 3 Report
The manuscript by Bodulev et al. used heterogeneous enzyme-linked oligonucleotide assay coupled with mismatched catalytic hairpin to quantify microRNA-155 in human cells. Although the data seem to be promising, the experimental design demands elaborate conditions, which should be strictly executed to guarantee that the results are reliable. The manuscript also referred to many publications containing crucial information to fully comprehend it. Although these citations are reasonable and appropriate, the manuscript should be able to stand alone itself, instead of demanding the readers to shuttle between several publications while reading it. To guarantee the reliability and plausibility of the results, I suggest that some points should be elucidated and addressed the in the manuscript.
1. The anti-FluAb should be (briefly or fully) described, despite citation of reference #19.
2. Why was 10mM Tris-HCl at pH 7.2 used as buffer, instead of the most popular Phosphate-based types? How does it impact the experimental data? Is there any advantage of using this buffer?
3. Experimental design in Figure 1: Theoretically, no matter how the target detection, there is still possibility of hybridization between Flu-HP1 and B-HP2, that is to say, B-HP2 still can hybridize with Flu-HP1 even the target was not detected. How should this problem be solved or minimized? Although Figure 5 partially relieved this concern, it should be thoroughly explained because it is critical to the results. What is the probability of the data in Figure 5 (is it statistically meaningful?) and the mechanisms lying behind to ensure that the subsequent amplification steps only happen on the target hybridization? They should be discussed in the interpretation of Figure 5 in section 3.4.
4. How were the equation in line 202 and in Figure S5 determined? Please clearly explain it in the manuscript.
5. How could the miRNA open the hairpin structure of the Flu-HP1 for hybridization? How could the B-HP2 displace the miRNA? What is the binding affinity between miRNA-Flu-HP1 and B-HP2-Flu-HP1, as well as the mechanisms of these processes? All of them should be (briefly or fully) mentioned in the manuscript.
6. The designed experimental procedure underwent several incubating/washing steps, which may interfere and deteriorate the detected signal. Is there any solution for this issue, especially to detect the target in real biological samples?
Round 2
Reviewer 3 Report
The manuscript can be published in its current form, since the questions noted in the review have been addressed in the revision.
Author Response
The reviewer wrote that "The manuscript can be published in its current form". Additional comments are absent.